# Genetic Variants in Genes Related to Lung Function and Interstitial Lung Diseases Are Associated with Worse Outcomes in Severe COVID-19 and Lung Performance in the Post-COVID-19 Condition

**DOI:** 10.3390/ijms26052046

**Published:** 2025-02-26

**Authors:** Ingrid Fricke-Galindo, Salvador García-Carmona, Brandon Bautista-Becerril, Gloria Pérez-Rubio, Ivette Buendia-Roldan, Leslie Chávez-Galán, Karol J. Nava-Quiroz, Jesús Alanis-Ponce, Juan M. Reséndiz-Hernández, Esther Blanco-Aguilar, Jessica I. Erives-Sedano, Yashohara Méndez-Velasco, Grecia E. Osuna-Espinoza, Fidel Salvador-Hernández, Rubén Segura-Castañeda, Uriel N. Solano-Candia, Ramcés Falfán-Valencia

**Affiliations:** 1HLA Laboratory, Instituto Nacional de Enfermedades Respiratorias Ismael Cosío Villegas, Mexico City 14080, Mexico; ingrid_fg@yahoo.com.mx (I.F.-G.); q.c.salvadorgc@gmail.com (S.G.-C.); brandon.bautistab@gmail.com (B.B.-B.); glofos@yahoo.com.mx (G.P.-R.); 14alanisponce@gmail.com (J.A.-P.); 2Translational Research Laboratory on Aging and Pulmonary Fibrosis, Instituto Nacional de Enfermedades Respiratorias Ismael Cosío Villegas, Mexico City 14080, Mexico; ivettebu@yahoo.com.mx; 3Laboratory of Integrative Immunology, Instituto Nacional de Enfermedades Respiratorias Ismael Cosío Villegas, Mexico City 14080, Mexico; lchavezgalan@gmail.com; 4Laboratorio Clínico, Centro Especializado de Atención a Personas con Discapacidad Visual, Instituto de Salud del Estado de México, Naucalpan 53000, Mexico State, Mexico; dr.jmresendiz@gmail.com; 5Facultad de Medicina Benemérita, Universidad Autónoma de Puebla, Puebla de Zaragoza 72420, Puebla, Mexico; esther.blancoag@gmail.com; 6Instituto de Ciencias Biomédicas, Universidad Autónoma de Ciudad Juárez, Ciudad Juárez 32310, Chihuahua, Mexico; al177671@alumnos.uacj.mx; 7Unidad Académica Profesional Chimalhuacán, Universidad Autónoma del Estado de México, Nezahualcóyotl 56353, Mexico State, Mexico; yashoharaand@gmail.com; 8Facultad de Ciencias Químico Biológicas, Universidad Autónoma de Sinaloa, Culiacán Rosales 80030, Sinaloa, Mexico; greciaosuna.fcqb@uas.edu.mx; 9Escuela Nacional de Medicina y Homeopatía, Instituto Politécnico Nacional, Mexico City 07320, Mexico; fidelenmh12@gmail.com; 10Facultad Interdisciplinaria de Ciencias Biológicas y de Salud, Universidad de Sonora, Hermosillo Sonora 83000, Sonora, Mexico; r.seguracast030800@gmail.com; 11Facultad de Biología, Universidad Michoacana de San Nicolás de Hidalgo, Morelia 58000, Michoacán, Mexico; 1415049d@umich.mx

**Keywords:** COVID-19, post-COVID-19, interstitial lung diseases, pulmonary function, invasive mechanical ventilation, FAM13A, DSP, TERT, THSD4

## Abstract

Genetic variants related to susceptibility to chronic respiratory conditions such as interstitial lung disease (ILD) could share critical pathways in the pathogenesis of COVID-19 and be implicated in COVID-19 outcomes and post-COVID-19. We aimed to identify the participation of genetic variants in lung function and ILD genes in severe COVID-19 outcomes and post-COVID-19 condition. We studied 936 hospitalized patients with COVID-19. The requirement of invasive mechanical ventilation (IMV) and the acute respiratory distress syndrome (ARDS) classification were considered. The mortality was assessed as the in-hospital death. The post-COVID-19 group included 102 patients evaluated for pulmonary function tests four times during the year after discharge. Five variants (*FAM13A* rs2609255, *DSP* rs2076295, *TOLLIP* rs111521887, *TERT* rs2736100, and *THSD4* rs872471) were genotyped using TaqMan assays. A multifactor dimensionality reduction method (MDR) was performed for epistasis estimation. The *TERT* rs2736100 and *THSD4* rs872471 variants were associated with differential risk for ARDS severity (moderate vs. severe, CC + CA, *p* = 0.044, OR = 0.66, 95% CI = 0.44–0.99; and GG *p* = 0.034, OR = 2.22, 95% CI = 1.04–4.72, respectively). These variants and *FAM13A* rs2609255 were also related to pulmonary function post-COVID-19. The MDR analysis showed differential epistasis and correlation of the genetic variants included in this study. The well-known variants in recognized genes related to pulmonary function worsening and interstitial disorders are related to the severity and mortality of COVID-19 and lung performance in the post-COVID-19 condition.

## 1. Introduction

Coronavirus disease 2019 (COVID-19) has created significant challenges worldwide, and our understanding of this disease continues to evolve [1]. Several genome-wide association studies have been performed in multiple ethnic groups, in which genes mainly related to the immune response were associated with the severity of COVID-19; for instance, inflammatory signaling (*JAK1*), immunometabolism (*SLC2A5* and *AK5*) [2], host antiviral defense mechanisms and mediators of inflammatory organ damage (*IFNAR2* and *TYK2*) [3], interferon signaling (*IL10RB* and *PLSCR1*), leucocyte differentiation (*BCL11A*), and blood-type antigen secretor status (*FUT2*) [4]. In addition, the first genome-wide significant association for the post-COVID-19 condition was for the FOXP4 locus, previously associated with COVID-19 severity, lung function, and cancers. Remarkably, the study has supported the role of pulmonary dysfunction in COVID-19 severity and in the development of long COVID or post-COVID-19 condition [5].

Single-Nucleotide Variants (SNVs) related to susceptibility to pulmonary function and respiratory diseases [6,7] could share critical pathways involved in the susceptibility and pathogenesis of COVID-19. These biological routes, including those involving the *FAM13A*, *DSP*, *TOLLIP*, *TERT* [8], and *THSD4* genes, have not been entirely studied in COVID-19 and post-COVID-19 condition. These genes have been previously related to lung function and susceptibility to chronic and respiratory lung diseases, such as chronic obstructive pulmonary diseases, lung cancer, interstitial lung disease, idiopathic pulmonary fibrosis, and mycobacterium tuberculosis [8,9,10,11,12].

*FAM13A* has been linked with an increased risk of COPD and is associated with β-catenin signaling. In SARS-CoV-2 infection, Wnt/β-catenin signaling affects immune cell function, suppresses the proliferation and self-renewal of alveolar macrophages, and promotes the production of pro-inflammatory mediators [13]. Moreover, FAM13A is involved in the TGF-β1-induced fibrotic response in the airway epithelium [14], which is recognized as the dominant chronic immune response in COVID-19 [15]. Meanwhile, desmoplakins (DSP) are critical components of desmosomes that maintain cardiac and respiratory epithelial integrity [16]. Mainly, the rs2076295 locus is directly responsible for differential RNA expression of DSP in primary epithelial cells [11], and the low levels of DSP in the lung cause increased expression of epithelial–mesenchymal transition and extracellular matrix genes promoting pulmonary fibrosis [17], one of the major concerns arising after the recovery from COVID-19 [18].

TOLLIP (toll-interacting protein) is essential to immune cell activation, cell survival, pathogen defense, and other biological processes related to inflammation and innate immune response, which are crucial in virus infections like COVID-19 [19]. Indeed, its deficiency results in ACE2 stabilization and elevated SARS-CoV-2 infection [20]. On the other hand, *THSD4* (thrombospondin type I domain-containing 4) has been positively correlated with inflammatory markers in patients with COVID-19 [21].

Finally, the *TERT* gene responsible for producing human telomerase reverse transcriptase (hTERT) is closely related to the viral RNA-dependent RNA polymerase (RdRP) [22] and linked to inadequate suppression of the innate immune response, increased risk of developing a cytokine storm, lung injury, and severe COVID-19 [23]. The rs2736100 variant has been related to the telomeres’ length [24], and it has been described that short telomeres limit the alveolar epithelial regenerative potential [25] and could be involved in the prognosis of patients recovered from COVID-19.

We hypothesize that SNVs in these lung function candidate genes are implicated in COVID-19 pathogenesis. This study aims to identify the association of *FAM13A*, *DSP*, *TOLLIP*, *TERT*, and *THSD4* variants with severe COVID-19 and post-COVID-19 condition in a Mexican mestizo population.

## 2. Results

### 2.1. Clinical and Demographical Characteristics

We included 936 subjects with COVID-19 diagnosis. The patients were classified according to survival (survivors and non-survivors), IMV requirement (IMV and non-IMV), and ARDS severity (mild, moderate, and severe). These groups’ clinical and demographic data are presented in Table 1 and Appendix A. The survival comparison presented higher differences than the other classifications. Older subjects with comorbidities (systemic arterial hypertension (SAH), cardiopathies, and preexisting chronic respiratory diseases (CRD, i.e., chronic obstructive pulmonary disease, obstructive sleep apnea syndrome, and/or asthma)) and lower levels of PaO_2_/FiO_2_ at hospital admission were more frequently in the non-survivors group, and most of the patients in this group required IMV during their hospital stay.

Meanwhile, patients requiring IMV were older, had a more extended hospital stay, and presented lower PaO_2_/FiO_2_ at hospital admission and higher mortality than patients in the non-IMV group. The frequencies of males, co-morbidities, and smokers were similar in both groups, although all were slightly higher in the IMV group (Appendix A).

For patients classified according to ARDS severity, differences were observed in the PaO_2_/FiO_2_ at hospital admission and outcomes related to the severity of the disease, such as days of hospital stay, mortality, and IMV requirement. The age and co-morbidities were not statistically significant between the mild, moderate, and severe groups (Table 1).

### 2.2. Genetic Association of FAM13A, TERT, DSP, TOLLIP, and THSD4 Variants with the Survival and Severity of Patients with COVID-19

The alleles and genotype frequencies were not statistically significantly different in the survival comparison. However, a marginal *p*-value was observed for the recessive model in the *DSP* rs2076295 variant. Then, the G allele of this variant was found to be associated with non-survival risk in the linear regression analysis adjusting for age, SAH, cardiopathies, CRD, PaO_2_/FiO_2_ at admission, and the IMV requirement (*p* = 0.047), even when age, PaO_2_/FiO_2_, and IMV requirement were only considered as co-variables (*p* = 0.039). Additionally, the TG genotype was found to have a low risk of death when the last three variables were included as co-variables in the analysis (*p* = 0.011, OR = 0.58, 95% CI = 0.39–0.88), as well as for the TT + TG genotypes (*p* = 0.021, OR = 0.63, 95% CI = 0.43–0.93). The remaining variants were not statistically significant when the logistic regression was performed (Table 2).

The frequencies were not statistically different in the IMV and non-IMV comparisons, even in the logistic regression model, which included age and PaO_2_/FiO_2_ at admission as co-variables. However, trending values were observed for the *THSD4* rs872471 variants (Appendix A).

In the analysis with the ARDS severity, the A allele of the *TERT* rs2736100 variant was observed as a low-risk factor when compared to the mild vs. moderate groups, and C was a risk allele when the moderate group was compared to the severe group. Likewise, in the genetic association models, the conjunction of CA + CC genotypes was observed as a low-risk factor for the moderate severity of ARDS. In contrast, the GG genotype of the *THSD4* rs872471 variant was more frequent in the severe ARDS group compared to the moderate ARDS group. Nevertheless, none of these variants remained associated when the logistic regression model was performed, adjusting for PaO_2_/FiO_2_ at admission, steroid administration, and the days since symptom onset (Table 3).

### 2.3. Genetic Association Study in the Post-COVID-19 Condition

One hundred and two patients were followed after COVID-19 discharge and included in the post-COVID-19 group. The patients were both male and female, had a median age of 56 years, were overweight, and around one-third of the group smoked and had co-morbidities such as T2DM and/or SAH. Most of the patients required IMV during their hospitalization (Table 2). More than half of the group (63.7%) were classified as having moderate ARDS during their hospitalization due to COVID-19, 13.7% as mild, and 22.5% as having severe ARDS.

Table 4 shows the values of the pulmonary function tests at each follow-up. The median value for each test was similar, and no statistical differences were observed between the four assessments.

Genetic variants were also determined in the post-COVID-19 group. The allele and genotype frequencies are shown in Appendix A. Overall, the frequencies were similar to those reported in the previous tables (Table 3, Appendix A) for the different COVID-19 groups.

We wondered if some of the studied variants were associated with susceptibility to the post-COVID-19 condition. Therefore, we compared the frequencies of the post-COVID-19 group with those of the survivors without post-COVID-19, but we did not find any significant differences.

We also evaluated whether there were differences in the values of the pulmonary function tests according to the genotypes of the studied variants. We performed a generalized linear model for each variable in the pulmonary function test at the four follow-ups (Table 4) as the dependent variable, and each studied genetic variant served as the independent variable (Table 1), adjusting for age and sex as covariates due to their known influence on pulmonary capacity [26,27].

The influence of genetic variants on pulmonary function test scores was observed at different times. In the 4th follow-up, the DL_CO_ (%) of TT subjects was different from that of the GG carriers of the *FAM13A* rs2609255, and the FEV1 (%) varied according to the *TERT* rs2736100 genotypes when the pulmonary function tests were performed for the first time at hospital discharge (Appendix A). Moreover, the influence of the DSP rs2076295 genotypes was observed in the FEV_1_/FVC values of the 3rd follow-up and the FEV_1_, FVC, and ∆SpO_2_ (%) after 6MWT after a year of the hospitalization discharge (4th follow-up, Appendix A). Finally, the herein-associated *THSD4* rs872471 variant was related to the meters reported by the 6MWT in the first two follow-ups and the FEV_1_ and the DL_CO_ values (both in percentages) of the last follow-up (Appendix A). Although significant differences were observed in the mentioned cases, we acknowledge the underpower of the analyses due to the small sample sizes observed mainly for the homozygous minor allele in some genetic variants. The remaining analyses of the different pulmonary function tests and genetic variants were not statistically significantly different.

### 2.4. Multifactor Dimensionality Reduction (MDR) Analyses

We looked for the influence of gene–gene interactions on clinical outcomes. The sample sizes were balanced using the under-sampling method. The best MDR model was selected when the values of testing accuracy were >0.55 and maximum cross-validation consistency was obtained [28]. The *TOLLIP* rs111521887 variant was not included in the analyses due to its low minor allele frequency.

The best MDR model for the mortality study had a testing accuracy of 0.57, 10/10 of cross-validation consistency, and included the three SNVs *DSP* rs2076295, *FAM13A* rs2609255, and *THSD4* rs872471. The distribution of cases and controls according to the different combinations is shown in Figure 1a. Some combinations could be observed as high-mortality-risk genotypes. The interaction graph showed a synergistic interaction between the *DSP* and *FAM13A* and between the *FAM13A* and the *THSD4* variants. Antagonism effects were observed for the interaction of *THSD4* with *DSP* and *TERT*. In contrast, no association was observed for *TERT* with *FAM13A* and *DSP* loci (Figure 1b).

Meanwhile, when the IMV requirement was evaluated, the best model had a testing accuracy of 0.56 (10/10 cross-validation) and included in the model the four assessed variants (Figure 2a). The interaction graph mainly revealed synergistic interactions (epistasis) (Figure 2b), with the strongest synergism for the *TERT* and *FAM13A* loci. Similarly, a powerful synergistic interaction was found between *TERT* and *FAM13A* when the severity of ARDS was compared in the analysis (moderate as controls and severe as cases; Figure 3a), but antagonism effects were observed for *DSP* with *TERT* and *FAM13A* loci. The best model for ARDS severity included the combination of the *TERT* and *FAM13A* variants and presented a testing accuracy of 0.66 (10/10 cross-validation) (Figure 3b).

## 3. Discussion

In our study investigating 936 hospitalized patients with COVID-19, we identified an association between a decreased risk of death and the TG heterozygous genotype and the recessive model (TT + TG genotypes) in *DSP* rs2076295. Recent research on COVID-19 highlights a relationship of the *DSP* gene with the lncRNA PIRAT (PU.1-induced regulator of alarmin transcription) that, during SARS-CoV-2 infection, drives the expression of alarmin S100A8 and S100A9, which together promote the differentiation of monocytes to macrophages and contribute to myeloid imbalances during severe COVID-19 [31]. In idiopathic pulmonary fibrosis (IPF), the G allele has been identified as the risk variant and associated with a decreased DSP RNA expression, resulting in a dysfunction in cell migration and the remodeling process in the lungs. In this sense, the protective effect observed with the common allele could be related to maintaining airway epithelial integrity [11,17]. However, the trade-off observed for IPF and COPD [17] can also be observed herein, since in the post-COVID-19 group, the highest pulmonary test scores were found for the patients with the GG genotype, suggesting differential activity of desmoplakins according to the timely disease process (acute or chronic) and the patient’s physiological status (i.e., age, comorbidities, and inflammatory status).

Meanwhile, *THSD4* has been studied multiple times in COPD [32]. Although little has been explored for COVID-19, we found that in the genetic association models, the GG genotype of the *THSD4* variant rs872471 was more frequent in the severe ARDS group. This relationship is likely seen in the interaction that the *THSD4* gene has with the metabolism of the family of extracellular calcium-binding thrombospondins involved in inflammation, apoptosis, coagulation, wound healing, and the interaction with extracellular components, receptors, and cytokines that could aggravate the disease [32]. COVID-19 patients had significantly higher levels of thrombospondins than healthy controls and were positively correlated with inflammatory markers, including ESR, CRP, PCT, ferritin, and biochemical parameters [21]. Moreover, the *THSD4* gene has been related to two chemokine genes (*CCL18* and *CXCL12*) and a TNF family receptor that participates in respiratory tract mucosal immunity and influences lung function, which is affected in moderate and severe cases of COVID-19 [33], as we found that it was related to the 6MWT, the % FEV_1_, and DL_CO_. Precisely, thrombospondin-1 has been correlated with % FEV_1_ in patients with COPD and is involved in activating the TGF-β pathway [34].

The A allele of the *TERT* rs2736100 variant was observed as a low-risk factor compared to the mild vs. moderate group, and C was observed as a risk allele when the moderate group was compared with the severe group. Recent research has pointed out the relationship between the *TERT* gene and the consistent presence of short telomeres in patients with severe COVID-19 symptoms [35]. As evidence grows that short telomeres increase the risk of severe COVID-19, several molecular mechanisms have been explained to address the connection between short telomeres and severe COVID-19 and suggest that T cell telomere length (TL) affects the adaptive immune response and the innate immune response to SARS-CoV-2 infection. Individuals with long T cell telomeres show a strong T cell response and a strong suppression of the innate immune response accompanied by moderate activity of the innate immune response. In contrast, individuals with short T cell telomeres show weak T cell response and inadequate suppression of the innate immune response accompanied by strong activity of the innate immune response, expressed as a storm of cytokines, lung injury, and severe COVID-19 [35]. Short telomeres were associated with an increased risk of critical illness, defined as admission to the intensive care unit (ICU) or non-ICU death. Finally, lung tissue from patients with very short telomeres shows signs of senescence in structural and immune cells [36].

Finally, SNVs in the *FAM13A* gene have been associated with an increased risk of COPD and IPF [8]; the most critical part of the FAM13A protein is its N-terminal extension containing the Rho-GAP domain involved in lung endothelial barrier function, which is often deregulated in lung diseases [9]. Patients with COVID-19 exhibit stronger humoral immunity, elevated Rho-GAP and mTOR pathway activities, and higher IFN-I signaling [37]. FAM13A is also associated with β-catenin signaling that is generally activated during injury repair and tissue regeneration [38]. In SARS-CoV-2 infection, Wnt/β-catenin signaling leads to the assembly of an “unconventional” β-catenin–HIF-1α complex that impairs immune cell function, suppresses proliferation and self-renewal of alveolar macrophages, and simultaneously promotes the production of pro-inflammatory mediators in vitro and in vivo [39]. In the post-COVID-19 group, we found that the T allele of rs2609255 variants was associated with lower diffusing capacity of the lungs for carbon monoxide, which agrees with previous studies including patients with IPF [40].

In summary, according to our findings, the involvement of *DSP*, *TERT*, and *THSD4* during the acute phase of COVID-19 is related to the inflammatory process, cell migration, cytokines release, and the adaptive and innate immune responses. Meanwhile, in the chronic manifestation (post-COVID-19 condition), the main problem is related to the limited regeneration of respiratory cells, as has been previously described [41], characterized by the participation of *FAM13A* in injury repair and tissue regeneration, *DSP* in the maintenance of airway epithelial integrity, *TERT* mediating cell proliferation, and the promotion of fibrotic mechanisms of *THSD4* via the TGF-β pathway.

Our study is not exempt from limitations. Firstly, it employed a single-center design and was conducted in a third-level hospital for respiratory diseases, which was focused on severe or critical COVID-19 during the pandemic; thus, patients with mild or asymptomatic disease were not included in this study, nor was a control group, since the diagnostic tests were prioritized for patients and healthcare professionals. Second, the sample collection period was extensive, potentially including different SARS-CoV-2 variants we did not consider. In addition, we did not follow up on mortality during the post-COVID-19 stage; therefore, the results are focused on survivor patients who have been discharged after one year since COVID-19 and in which there could also be follow-up attrition rates. However, our results provide some perspectives on the influence of SNVs in genes related to ILDs, respiratory traits, and severe COVID-19 outcomes.

## 4. Materials and Methods

### 4.1. Subjects Included

Patients were recruited in the Instituto Nacional de Enfermedades Respiratorias Ismael Cosio Villegas (INER) at Mexico City, Mexico, a third-level referral center for respiratory disease, where severe cases of COVID-19 were attended. We evaluated two groups of patients with a confirmed diagnosis of COVID-19 through an RT-PCR SARS-CoV-2 test (Appendix A); the first included those hospitalized in the INER, consecutively enrolled from August 2020 to December 2022, and who were >18 years old. Patients were excluded from the analyses when their information (outcome, clinical, and demographic variables) was unavailable. This group was further classified according to three clinical outcomes: (1) non-survival, evaluated as in-hospital deceased; (2) requirement of invasive mechanical ventilation (IMV); and (3) severity of the acute respiratory distress syndrome (ARDS) according to PaO_2_/FiO_2_ upon hospital admission (mild > 200, moderate 100–200, and severe < 100). Only <1.0% of the patients included in this study were administered at least one dose of the vaccine; therefore, this information was not considered in the analyses.

The second group comprised patients with a post-COVID-19 condition followed up for 12 months (four follow-ups every three months) after discharge from severe COVID-19 in the INER. All of the patients in this group were diagnosed with severe COVID-19 when hospitalized (April 2020 to August 2021) and required respiratory support through IMV or high-flow nasal cannula oxygen therapy. Fifty-seven subjects from this group were also included in the survivors group. Therefore, they were only included in the post-COVID-19 group for the survivors vs. post-COVID-19 study. They showed pulmonary dysfunction determined by a decrease in their forced vital capacity or desaturation in the 6 min walking test (6MWT) and/or interstitial thickening in the computed tomography, as previously reported [42,43]. The tests performed in each follow-up were spirometry, DLCO, and 6MWT.

### 4.2. Genotyping

Genomic DNA was isolated from blood cells using standard techniques. The *FAM13A* rs2609255, *TERT* rs2736100, *DSP* rs2076295, *TOLLIP* rs111521887, and *THSD4* rs872471 variants were genotyped using TaqMan ^TM^ assays. These intron SNVs are found in genes selected according to their relevance in pulmonary diseases and lung function. These variants affect regulatory motifs and the expression evaluated from eQTL studies (Table 5) according to HaploReg v4.2 [44].

### 4.3. Statistical Analysis

Categorical data are presented as absolute numbers (frequency in percentage). Continuous data are described as medians (interquartile range, Q_1_–Q_3_). The normality distribution of the data was assessed with the Shapiro–Wilk and/or Kolmogorov–Smirnov tests, as appropriate. χ2 test and Mann–Whitney U or Kruskal–Wallis tests were used to compare variables between groups. The genetic association study was assessed in PLINK v1.07 [45], and a logistic regression model was performed to adjust for those co-variables that had statistical differences between the comparison groups (see Table 1). The evaluation of the pulmonary function test scores according to genotype and co-variable adjustment was evaluated using a generalized linear model. The statistical tests were conducted in RStudio v4.1.2 [46]. A *p*-value < 0.05 was set as statistically significant in all cases.

In addition, we evaluated gene–gene interactions using the non-parametric method multifactor dimensionality reduction (MDR) v3.0.2 [29,30]. Three analyses were performed for each evaluated phenotype (mortality, IMV requirement, and moderate vs. severe ARDS). The comparison groups were unbalanced; therefore, we performed under-sampling, which involves randomly removing members of the over-represented group from the dataset until it is balanced. Subjects with missing values were not included. All procedures were performed using ten-fold cross-validation. The selection criteria for a final candidate model were based on the best cross-validation consistency, training accuracy, and testing accuracy (>0.55). A permutation test determined the statistical significance of the training and testing accuracy for the best candidate model.

## 5. Conclusions

The well-known variants in recognized genes related to pulmonary function worsening (*THSD4* rs872471) and interstitial disorders (*FAM13A* rs2609255, *DSP* rs2076295, *TERT* rs2736100) are related to the severity and mortality of COVID-19 and lung performance in the post-COVID-19 condition.

## Figures and Tables

**Figure 1 ijms-26-02046-f001:**
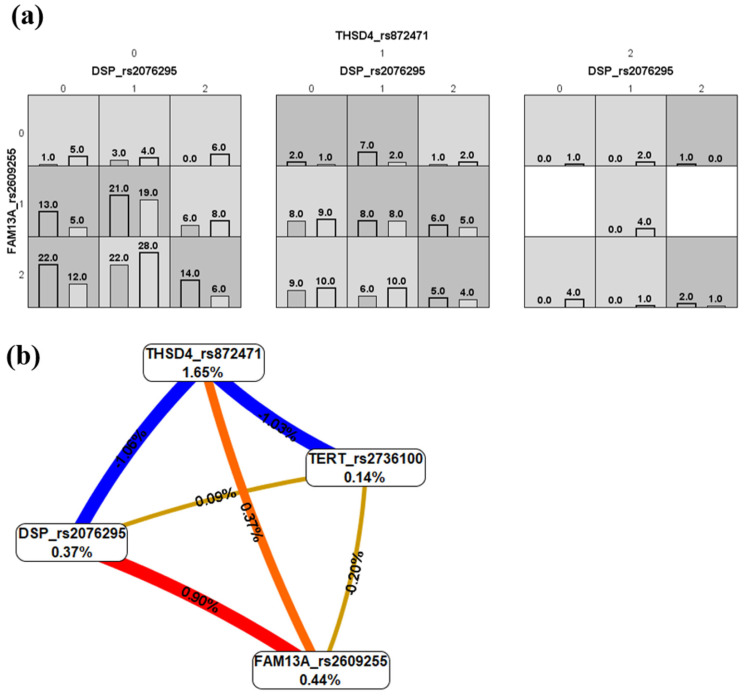
Multifactor dimensionality reduction analysis for mortality in COVID-19. (**a**) Bar charts showing the distributions of the genotypes’ combinations, including the *DSP* rs2076295, *FAM13A* rs2609255, and *THSD4* rs872471 variants (for each graph, the left bars correspond to cases, and right bars to controls. 0, homozygous for the common allele; 1, heterozygous; 2, homozygous for the uncommon allele). High-risk genotypes can be observed in dark grey, and low-risk genotypes in light grey. (**b**) Gene–gene interaction graph. The red and orange colors indicate strong and moderate synergism; gold denotes no association or independence of selected loci effects; and blue indicates strong antagonism [29,30].

**Figure 2 ijms-26-02046-f002:**
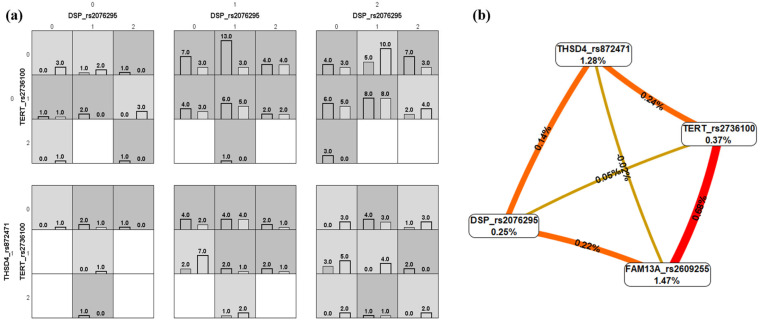
Multifactor dimensionality reduction analysis for invasive mechanical ventilation requirement. (**a**) Bar charts showing the distributions of the genotypes’ combinations of the genetic variants included in this study (for each graph, the left bars correspond to cases, and the right bars to controls. 0, homozygous for the common allele; 1, heterozygous; 2, homozygous for the uncommon allele). High-risk genotypes can be observed in dark grey, and low-risk genotypes in light grey. (**b**) Gene–gene interaction graph. The red and orange colors indicate strong and moderate synergism; gold denotes no association or independence of selected loci effects [29,30].

**Figure 3 ijms-26-02046-f003:**
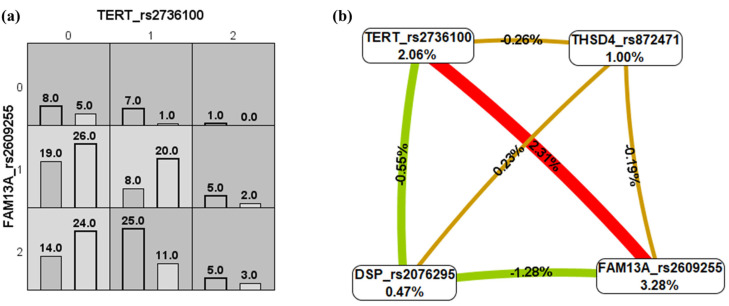
Multifactor dimensionality reduction analysis for acute respiratory distress syndrome severity in COVID-19 (moderate vs. severe). (**a**) Bar charts showing the distributions of the genotype combinations of the genetic variants included in this study (for each graph, the left bars correspond to cases, and the right bars to controls. 0, homozygous for the common allele; 1, heterozygous; 2, homozygous for the uncommon allele). High-risk genotypes can be observed in dark grey, and low-risk genotypes in light grey. (**b**) Gene–gene interaction graph. The red color indicates strong and moderate synergism; gold denotes no association or independence of selected loci effects; and green indicates moderate antagonism [29,30].

**Table 1 ijms-26-02046-t001:** Clinical and demographic data of patients with severe COVID-19 according to survival status and acute respiratory distress syndrome (ARDS) severity.

Variable	Non-Survivors (n = 369)	Survivors (n = 554)	*p*-Value	Mild (n = 183)	Moderate (n = 335)	Severe (n = 398)	*p*-Value	Post-COVID-19 (n = 102)	*p*-Value ^g^
Age, years	63.5 (55–73)	56 (48–64)	**<0.001**	59 (51–67)	59 (49–68)	59 (51–69)	0.686	56 (49–64)	0.862
Males, n (%)	266 (72.1)	371 (67.0)	0.094	130 (71.0)	225 (67.2)	284 (71.4)	0.41	39 (38.2)	**<0.001**
BMI, kg/m^2^	29.2 (25.7–33.4)	29.9 (26.6–33.4)	0.089	29.2 (25.6–32.9)	29.4 (26.1–33.3)	29.7 (26.2–33.8)	0.442	28 (26–31)	**0.003**
Hospital stay, days	19 (12–29)	20 (13–32)	0.176	15 (9–23)	21 (12–31)	21 (15–34)	**<0.001 ^a^**	27 (17–35)	**<0.001**
Symptoms onset, days	9 (7–14)	9 (7–13)	0.907	9 (7–12)	9 (7–13)	10 (7–14)	**0.003 ^b^**	9 (7–12)	0.338
Comorbidities									
T2DM, n (%)	116 (31.4)	149 (26.9)	0.137	49 (26.8)	103 (30.7)	106 (26.6)	0.441	32 (31.4)	0.328
SAH, n (%)	144 (39.0)	173 (31.2)	**0.013**	54 (29.5)	122 (36.4)	140 (35.2)	0.26	27 (26.5)	0.409
Cardiopathies, n (%)	25 (6.8)	14 (2.5)	**0.002**	9 (4.9)	11 (3.3)	18 (4.5)	0.588	5 (4.9)	0.105
CRD, n (%)	44 (11.9)	38 (6.8)	**0.009**	19 (10.4)	22 (6.6)	41 (10.3)	0.157	16 (15.7)	**0.001**
Smoking, n (%)	114 (30.9)	148 (26.7)	0.18	51 (27.9)	88 (26.3)	122 (30.6)	0.429	30 (29.4)	0.626
TI, packs/year	5.2 (1.7–20.0)	4.6 (1.5–15.0)	0.419	4.8 (1.7–17.7)	4.2 (1.5–12.0)	5.2 (1.5–20)	0.513	4 (1–10)	0.646
Steroid administration n (%)	317 (85.9)	448 (80.9)	0.050	156 (85.2)	262 (78.2)	345 (86.7)	**0.003 ^c^**	56 (54.9)	**<0.001**
PaO_2_/FiO_2_ at hospital admission	92 (68–152)	137 (85–203)	**<0.001**	227 (211–252)	147 (123–174)	73 (61–87)	**<0.001 ^d^**	151 (102–186)	0.326
IMV, n (%)	344 (93.2)	373 (67.3)	**<0.001**	93 (50.8)	270 (80.6)	355 (89.2)	**<0.001 ^e^**	96 (94.1)	**<0.001**
IMV, days	19 (12–28)	16 (10–28)	0.053	15 (10–22)	18 (12–29)	18 (12–28)	0.062	17 (9–24)	0.903
Deceased, n (%)	369 (100.0)	0 (0.0)	**<0.001**	46 (25.1)	121 (36.1)	199 (50.0)	**<0.001 ^f^**	0	1.000

Continuous data are presented as the medians (interquartile range, Q_1_–Q_3_), and categorical data as the absolute values (frequency in percentage). The statistical comparisons between tests were made using Mann–Whitney and Kruskal–Wallis tests with the Benjamini–Hochberg *p*-value correction method and Exact Fisher’s test. BMI, body mass index; CRD, preexisting chronic respiratory diseases (i.e., chronic obstructive pulmonary disease, obstructive sleep apnea syndrome, and/or asthma); IMV, invasive mechanical ventilation; PaO_2_/FiO_2_, ratio of the partial pressure of oxygen in arterial blood to the fraction of inspired oxygen; SAH, systemic arterial hypertension; T2DM, type 2 diabetes mellitus; TI, tobacco index. ^a^ Mild vs. moderate, *p* < 0.001, mild vs. severe, *p* < 0.001; ^b^ mild vs. severe, *p* = 0.01, moderate vs. severe, *p* = 0.01; ^c^ mild vs. moderate, *p* = 0.052, moderate vs. severe, *p* = 0.003; ^d^ mild vs. moderate, mild vs. severe, moderate vs. severe, *p* < 0.001 in all cases; ^e^ mild vs. moderate, mild vs. severe, moderate vs. severe, *p* < 0.001 in all cases; ^f^ mild vs. moderate, *p* = 0.014, mild vs. severe and moderate vs. severe, *p* < 0.001 in both cases; ^g^ post-COVID-19 vs. survivors (n = 497). The bold text indicates a significant statistical *p*-value.

**Table 2 ijms-26-02046-t002:** Genetic association study with non-survival risk in patients with severe COVID-19.

Gene/ Genetic Variant	Allele/Genotype	Non-Survivors	Survivors	*p*-Value	Adjusted *p*-Value ^a^
*FAM13A*		n = 347 (%)	n = 540 (%)		
rs2609255	T	489 (0.705)	729 (0.675)	0.189	0.313
	G	205 (0.295)	351 (0.325)		
	TT	172 (0.496)	245 (0.454)	0.419	0.331
	TG	145 (0.418)	239 (0.443)		0.134
	GG	30 (0.086)	56 (0.104)		
	Genetic association models				
	TT	172 (0.496)	245 (0.454)	0.222	0.573
	TG + GG	175 (0.504)	295 (0.547)		
	TT + TG	317 (0.914)	484 (0.897)	0.397	0.109
	GG	30 (0.086)	56 (0.104)		
*TERT*		n = 190 (%)	n = 352 (%)		
rs2736100	A	269 (0.708)	510 (0.724)	0.563	0.498
	C	111 (0.292)	194 (0.276)		
	AA	98 (0.516)	182 (0.517)		
	CA	73 (0.384)	146 (0.415)	0.394	0.919
	CC	19 (0.100)	24 (0.068)		0.247
	Genetic association models				
	AA	98 (0.516)	182 (0.517)	0.978	0.629
	CC + CA	92 (0.484)	170 (0.483)		
	CA + AA	171 (0.900)	328 (0.932)	0.191	0.243
	CC	19 (0.100)	24 (0.068)		
*DSP*		n = 352 (%)	n = 528 (%)		
rs2076295	T	396 (0.563)	620 (0.587)	0.306	**0.047**
	G	308 (0.437)	436 (0.413)		
	TT	118 (0.335)	178 (0.337)	0.173	0.211
	TG	160 (0.455)	264 (0.500)		**0.023**
	GG	74 (0.210)	86 (0.163)		
	Genetic association models				
	TT	118 (0.335)	178 (0.337)	0.953	0.655
	TG + GG	234 (0.664)	350 (0.663)		
	TT + TG	278 (0.790)	442 (0.837)	0.074	**0.046**
	GG	74 (0.210)	86 (0.163)		
*TOLLIP*		n = 176 (%)	n = 471 (%)		
rs111521887	C	337 (0.957)	911 (0.967)	0.401	0.598
	G	15 (0.043)	31 (0.033)		
	CC	161 (0.915)	441 (0.936)	0.475	0.987
	CG	15 (0.085)	29 (0.062)		0.316
	GG	0 (0.000)	1 (0.002)		
	Genetic association models				
	CC	161 (0.915)	441 (0.936)	0.433	0.329
	CG + GG	15 (0.085)	30 (0.064)		
	GG	176 (1.000)	470 (0.998)	1.000	0.989
	CC + CG	0 (0.000)	1 (0.002)		
*THSD4*		n = 320 (%)	n = 466 (%)		
rs872471	A	506 (0.791)	720 (0.773)	0.395	0.991
	G	134 (0.209)	212 (0.227)		
	AA	202 (0.631)	285 (0.612)	0.609	
	AG	102 (0.319)	150 (0.322)		0.94
	GG	16 (0.050)	31 (0.066)		0.893
	Genetic association models				
	AA	202 (0.631)	285 (0.612)	0.577	0.981
	AG + GG	118 (0.369)	181 (0.388)		
	AA + AG	304 (0.950)	435 (0.934)	0.337	0.882
	GG	16 (0.050)	31 (0.066)		

^a^ Logistic regression model with adjustment for the co-variables of age, systemic arterial hypertension, cardiopathies, pre-existing chronic respiratory diseases, PaO_2_/FiO_2_ levels at admission, and invasive mechanical ventilation requirement. Genotyping was performed according to sample availability; thus, the n genotyped is specified for each case. The bold text indicates a significant statistical *p*-value.

**Table 3 ijms-26-02046-t003:** Genetic association study with acute respiratory distress syndrome severity in patients with severe COVID-19.

Gene/ Genetic Variant	Allele/ Genotype	Mild	Moderate	Severe	*p*-Value	OR (95% CI)
*FAM13A*		n = 178	n = 321	n = 380		
rs2609255	T	246 (0.691)	439 (0.684)	521 (0.686)	NS	-
	G	110 (0.309)	203 (0.316)	239 (0.314)		
	TT	89 (0.500)	144 (0.449)	181 (0.476)	NS	-
	TG	68 (0.382)	151 (0.470)	159 (0.418)		
	GG	21 (0.118)	26 (0.081)	40 (0.105)		
	Genetic association models					
	TT	89 (0.500)	144 (0.449)	181 (0.476)	NS	-
	TG + GG	89 (0.500)	177 (0.551)	199 (0.524)		
	TT + TG	157 (0.882)	295 (0.754)	340 (0.895)	NS	-
	GG	21 (0.118)	26 (0.066)	40 (0.105)		
*TERT*		n = 140	n = 278	n = 108		
rs2736100	A	193 (0.689)	420 (0.755)	147 (0.680)	**0.041 ^a^**	**^a^ 0.72 (0.52–0.99)**
	C	87 (0.311)	136 (0.245)	69 (0.319)	**0.034 ^b^**	**^b^ 1.45 (1.03–2.05)**
	AA	65 (0.464)	158 (0.568)	51 (0.472)	NS	-
	CA	63 (0.450)	104 (0.374)	45 (0.417)		
	CC	12 (0.086)	16 (0.057)	12 (0.111)		
	Genetic association models					
	AA	65 (0.464)	158 (0.568)	51 (0.472)		
	CC + CA	75 (0.536)	120 (0.432)	57 (0.528)	**0.044 ^a^**	**^a^ 0.66 (0.44–0.99)**
	CA + AA	12 (0.086)	16 (0.057)	12 (0.111)	NS	-
	CC	128 (0.914)	262 (0.942)	96 (0.889)		
*DSP*		n = 180	n = 317	n = 379		
rs2076295	T	203 (0.564)	366 (0.577)	445 (0.587)	NS	-
	G	157 (0.436)	268 (0.423)	313 (0.413)		
	TT	56 (0.311)	104 (0.328)	135 (0.356)	NS	-
	TG	91 (0.505)	158 (0.498)	175 (0.462)		
	GG	33 (0.183)	55 (0.173)	69 (0.182)		
	Genetic association models					
	TT	56 (0.311)	104 (0.328)	135 (0.356)	NS	-
	TG + GG	124 (0.689)	213 (0.672)	244 (0.644)		
	TT + TG	147 (0.817)	262 (0.826)	310 (0.824)	NS	-
	GG	33 (0.183)	55 (0.173)	69 (0.182)		
*TOLLIP*		n = 176	n = 335	n = 88		
rs111521887	C	340 (0.966)	651 (0.972)	168 (0.954)	NS	-
	G	12 (0.034)	19 (0.028)	8 (0.045)		
	CC	164 (0.932)	317 (0.946)	80 (0.909)	NS	-
	CG	12 (0.068)	17 (0.051)	8 (0.091)		
	GG	0 (0.000)	1 (0.003)	0 (0.000)		
	Genetic association models					
	CC	164 (0.932)	317 (0.946)	80 (0.909)	NS	-
	CG + GG	12 (0.068)	18 (0.054)	8 (0.091)		
	CC + CG	176 (1.000)	334 (0.997)	88 (1.000)	NS	-
	GG	0 (0.000)	1 (0.003)	0 (0.000)		
*THSD4*		n = 157	n = 293	n = 330		
rs872471	A	249 (0.793)	471 (0.804)	502 (0.761)	NS	-
	G	65 (0.207)	115 (0.196)	158 (0.239)		
	AA	102 (0.650)	188 (0.642)	196 (0.594)	NS	-
	AG	45 (0.287)	95 (0.324)	110 (0.333)		
	GG	10 (0.064)	10 (0.034)	24 (0.073)		
	Genetic association models					
	AA	102 (0.650)	188 (0.642)	196 (0.594)	NS	-
	AG + GG	55 (0.350)	105 (0.358)	134 (0.406)		
	AA + AG	147 (0.936)	283 (0.966)	306 (0.918)	**0.034 ^b^**	**2.22 (1.04–4.72)**
	GG	10 (0.064)	10 (0.034)	24 (0.073)		

^a^ Mild vs. moderate; ^b^ moderate vs. severe. CI, confidence interval; NS, not statistically significant when comparing mild vs. moderate, mild vs. severe, or moderate vs severe; OR, odds ratio. Genotyping was performed according to sample availability; thus, the n genotyped is specified for each case. The bold text indicates a significant statistical *p*-value.

**Table 4 ijms-26-02046-t004:** Follow-ups of the pulmonary function tests of patients in the post-COVID-19 condition group.

Pulmonary Function Test	First	Second	Third	Fourth	*p*-Value ^a^
FEV_1_, %	93 (83–103)	94 (83–107)	97 (85–108)	97 (84–106)	0.816
FVC, %	86 (73–99)	90 (80–101)	89 (83–102)	91 (81–99)	0.940
FEV_1_/FVC	84 (78–90)	84 (80–89)	83 (80–88)	84 (80–88)	0.416
DL_CO_, %	79 (67–97)	83 (69–97)	82 (71–89)	76 (69–85)	0.384
6MWT, m	420 (131–495)	455 (360–503)	467 (408–530)	472 (386–530)	0.503
∆SpO_2_, %—6MWT	1 (0–2)	4 (3–7)	3 (2–5)	4 (2–7)	0.152

Continuous data are presented as median (interquartile range Q1–Q3). ^a^ Kruskal–Wallis test. FEV1 forced expiratory volume in the first second; FVC forced vital capacity; DLCO, diffusing capacity of the lungs for carbon monoxide; 6MWT, six-minute walk test; ∆SpO2, delta of oxygen saturation determination at onset and at the end of the performance of the 6MWT.

**Table 5 ijms-26-02046-t005:** Characteristics of the variants included in this study.

Genetic Variants	Motifs Changed	NHGRI/EBI GWAS Hits	GRASP QTL Hits	Selected eQTL Hits	TaqMan Id Assay
*FAM13A* rs2609255	Four altered motifs	One hit	One hit	-	C__15906608_10
*TERT* rs2736100	Foxa	Fifteen hits	Three hits	-	C___1844009_10
*DSP* rs2076295	Five altered motifs	One hit	-	Five hits	C__16167921_10
*TOLLIP* rs111521887	-	-	-	One hit	C_152324875_10
*THSD4* rs872471	HNF1, Hbp1, Nrf1	-	-	One hit	C____262062_10

GRASP, Genome-Wide Repository of Associations Between SNPs and Phenotypes; Foxa, Forkhead box class A; GWAS, genome-wide association study; Hbp1, HMG box protein 1; HNF1, hepatocyte nuclear factor 1; NHGRI/EBI, National Human Genome Research Institute/European Bioinformatics Institute; Nrf1, nuclear respiratory factor 1; QTL, quantitative trait loci.

## Data Availability

The datasets presented in this study can be found in online repositories. The name of the repository and accession numbers can be found below: Clinvar accession numbers: SCV005044895, SCV005044896, SCV005044897, SCV005044898, and SCV005044899. Organization ID: 507267. In https://www.ncbi.nlm.nih.gov/clinvar/. (accessed on 13 January 2025).

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
