# Peer review of "Genetic Variants in Genes Related to Lung Function and Interstitial Lung Diseases Are Associated with Worse Outcomes in Severe COVID-19 and Lung Performance in the Post-COVID-19 Condition"

_ijms, 2025, doi:10.3390/ijms26052046_

Round 1

Reviewer 1 Report

Comments and Suggestions for Authors

Dear Editor,

Dear Author,

I read and reviewed the very long paper submitted to International journal of molecular sciences.

Genetic variants in genes related to lung function and interstitial lung diseases are associated with worse outcomes in severe COVID-19 and lung performance in the post-COVID-19 condition.

It is an interesting report. It contains important information and it well conducted and described. Therefore, it should be published.

I have some comments:

1. Delete in the abstract the sentence: Four of five variants were found to be associated (line 38). Is confusing!

2. Please delete the first 12 lines in the discussion. Start the discussion as follows: In our study investigating 936 hospitalized patients with COVID-19 we identified……   

3. I would go deeper in the discussion explaining the difference between the acute COVID-19 and Post-COVID-19 for the identified gene variants.    

4. Some abbreviations are not explained in the tables. Spell out or explain! The tables should be readable even they are reported alone.     

5. Did patients receive therapeutics in the acute Phase? If yes, I would consider this aspect also in the analysis.  

Author Response

I read and reviewed the very long paper submitted to International journal of molecular sciences.

Genetic variants in genes related to lung function and interstitial lung diseases are associated with worse outcomes in severe COVID-19 and lung performance in the post-COVID-19 condition.

It is an interesting report. It contains important information and it well conducted and described. Therefore, it should be published.

 Authors’ response: We thank the Reviewer for the valuable comments to our manuscript.

I have some comments:

  1. Delete in the abstract the sentence: Four of five variants were found to be associated (line 38). Is confusing!

Authors’ response: We have deleted the sentence to a better comprehension of the study.

  1. Please delete the first 12 lines in the discussion. Start the discussion as follows: In our study investigating 936 hospitalized patients with COVID-19 we identified……   

Authors’ response: We removed the first and second paragraphs of the Discussion as suggested.

  1. I would go deeper in the discussion explaining the difference between the acute COVID-19 and Post-COVID-19 for the identified gene variants.    

Authors’ response: We thank for the Reviewer’s suggestion. We included this interesting point in the Discussion section (lines 287-290 and 337-344).

  1. Some abbreviations are not explained in the tables. Spell out or explain! The tables should be readable even they are reported alone.   

Authors’ response: We apologize for our oversight. We revised the abbreviations of the tables and included explanations for those missing.

  1. Did patients receive therapeutics in the acute Phase? If yes, I would consider this aspect also in the analysis.  

Authors’ response: We agree with the Reviewer’s comment. The treatment of hospitalized patients with COVID-19 was mainly based on the use of steroids. The availability of immunomodulators and antiviral drugs was strongly limited in our population. The use of steroids is included in Table 1, and it was considered a co-variable in the logistic regression model when the frequency of drug use was different between groups.

Reviewer 2 Report

Comments and Suggestions for Authors

This is a prospective study that included patients that were hospitalized with severe COVID-19 infection in a single center. Two group of patients with a confirmed diagnosis of COVID-19 were studied. The first group inluded patients that were evaluated during their hospitalization and the hospitalization outcome was studied, and the second group, that included patients with severe COVID-19 and  respiratory support through IMV or high-flow nasal cannula oxygen therapy during thier hospitalization was followed up for 12 months. The aim of the study was to identify the participation of genetic variants in lung function and ILD genes in the severe COVID-19 outcome and post-COVID-19 condition. 

Very interesting study.

  • Please provide a flow chart to better depict the methodology.
  • Improve clarity of figures, especially for bar charts and gene-gene interaction graphs, percentages are not clearly visible.
  • In the comorbidities of the studied population "chronic respiratory diseases" is mentioned, but it is not specified which (IPF, asthma, COPD, bronchiectasis, etc).
  • Where patients excluded? Provide detailed descriptions of patient inclusion/exclusion criteria and the rationale for their selection.
  • There is a lack of control group to test the validity of the results. Please, address limitations more comprehensively, including lack of control group, single-center design, and follow-up attrition rates.

Author Response

This is a prospective study that included patients that were hospitalized with severe COVID-19 infection in a single center. Two group of patients with a confirmed diagnosis of COVID-19 were studied. The first group inluded patients that were evaluated during their hospitalization and the hospitalization outcome was studied, and the second group, that included patients with severe COVID-19 and  respiratory support through IMV or high-flow nasal cannula oxygen therapy during thier hospitalization was followed up for 12 months. The aim of the study was to identify the participation of genetic variants in lung function and ILD genes in the severe COVID-19 outcome and post-COVID-19 condition. 

Very interesting study.

 Authors’ response: We thank the Reviewer for the valuable comments to our manuscript.

  • Please provide a flow chart to better depict the methodology.

Authors’ response: A flow chart with the methodology was included in Supplementary Figure 3.

  • Improve clarity of figures, especially for bar charts and gene-gene interaction graphs, percentages are not clearly visible.

Authors’ response: We thank the Reviewer’s comments. We have separated Figure 2 to make the bar charts and gene-gene interaction graphs more visible.

  • In the comorbidities of the studied population "chronic respiratory diseases" is mentioned, but it is not specified which (IPF, asthma, COPD, bronchiectasis, etc).

Authors’ response: We thank the Reviewer for the comment. The previous respiratory diseases were COPD, asthma, and obstructive sleep apnoea syndrome. We added the information in line 107 and Table 1.

  • Where patients excluded? Provide detailed descriptions of patient inclusion/exclusion criteria and the rationale for their selection.

Authors’ response: We have detailed the selection criteria in the Methods Section, lines 359 - 364.

  • There is a lack of control group to test the validity of the results. Please, address limitations more comprehensively, including lack of control group, single-center design, and follow-up attrition rates.

Authors’ response: We acknowledge these limitations, and they were added to the Limitations section as suggested (Line 345-353).

Round 2

Reviewer 1 Report

Comments and Suggestions for Authors

The authors followed the recomadations.

In my opinion the manuscript is ready.

Comments on the Quality of English Language

The authors followed the recomadations.

In my opinion the manuscript is ready.